# Mechanosensory neurons control sweet sensing in *Drosophila*

Yong Taek Jeong[1], Soo Min Oh[1], Jaewon Shim[2], Jeong Taeg Seo[1], Jae Young Kwon[3] & Seok Jun Moon[1]

Animals discriminate nutritious food from toxic substances using their sense of taste. Since taste perception requires taste receptor cells to come into contact with water-soluble chemicals, it is a form of contact chemosensation. Concurrent with that contact, mechanosensitive cells detect the texture of food and also contribute to the regulation of feeding. Little is known, however, about the extent to which chemosensitive and mechanosensitive circuits interact. Here, we show *Drosophila* prefers soft food at the expense of sweetness and that this preference requires labellar mechanosensory neurons (MNs) and the mechanosensory channel Nanchung. Activation of these labellar MNs causes GABAergic inhibition of sweet-sensing gustatory receptor neurons, reducing the perceived intensity of a sweet stimulus. These findings expand our understanding of the ways different sensory modalities cooperate to shape animal behaviour.

[1] Department of Oral Biology, BK21 PLUS Project, Yonsei University College of Dentistry, 50-1 Yonsei-ro Seodaemun-gu, Seoul 03722, Korea. [2] Division of Creative Food Science for Health, Korea Food Research Institute, Seongnam-si, Gyeonggi-do 13539, Korea. [3] Department of Biological Sciences, Sungkyunkwan University, Suwon 16419, Korea. Correspondence and requests for materials should be addressed to S.J.M. (email: sjmoon@yuhs.ac).

nimals must eat to survive, but not all food sources are equally desirable. Animals use their sense of taste to discriminate nutritious foods and toxic substances[1]. Although a food's taste is a major determinant of its acceptability, animals must assess a food's visual appearance, smell, temperature and texture as well. What we humans call 'flavour' is actually a complex multisensory picture of a food's general desirability[2]. In fact, we all have direct experience with the interaction of multiple sensory modalities in the general perception of food quality. Who hasn't noticed a change in a food's flavour on catching a cold severe enough to block their sense of smell?

Despite its obvious importance, the mechanisms by which multimodal sensory information is incorporated into feeding decisions are not well understood. Psychologists and neuroscientists have begun to explore the ways the individual channels of sensory input affect the perception of flavour, but our understanding of cross-modal interactions lags behind. This is partly due to difficulties with parsing the individual components that make up the gestalt of flavour perception, and partly due to technical difficulties associated with the controlled delivery of precisely defined multimodal stimuli. Because of these difficulties, we suggest the exploration of simpler model systems can help extend our understanding of the multisensory perception of flavour that directs feeding decisions.

In particular, we are interested in the ways neural circuits integrate taste and texture information. Texture is a product of mechanosensation. Animals, of course, use mechanosensory information to help determine their food's precise location[3], but it is the food's physical properties (for example, its hardness or viscosity) that contribute to determining its palatability. Several studies have demonstrated flavour perception can be altered by a food's hardness or viscosity[4–6]. In particular, Hollowood et al.[4] found in a group of human volunteers a negative correlation between food viscosity and perceived sweetness; as a food's viscosity increases, we perceive it as being less sweet. Since these sorts of interactions exist, they presumably offer some utility, but the neural mechanisms by which they help coordinate appropriate feeding behaviours are not understood in any system.

Drosophila present an especially attractive system for exploring interactions between taste and mechanosensation with regard to feeding decisions. Although both taste and olfaction are forms of chemosensation, because odorants are airborne and tastants are water-soluble, only taste requires contact with the stimulus. Indeed, while Drosophila olfactory sensilla lack mechanosensory neurons (MNs)[7], the gustatory receptor neurons (GRNs) of each taste sensillum are accompanied by a MN[8]. Thus, as a fly feeds

the sensory sensilla on its labellum (mouthparts) unavoidably receive concurrent taste and mechanical activation. In addition, the molecular genetic tools available in the fly allow us to examine the role each type of sensory information plays in directing feeding behaviour via selective activation or inactivation of each class of sensory neuron.

Here, we report our exploration of the circuit-level interactions between the perception of gustatory and mechanical stimuli that help direct feeding decisions in Drosophila. We have discovered Drosophila prefer soft food at the expense of sweetness and that this preference depends on labellar MNs and their expression of the mechanosensory channel Nanchung. Activation of these labellar MNs attenuates the perceived intensity of a sweet stimulus by suppressing the calcium responses of sweet GRN termini via the inhibitory neurotransmitter GABA. These findings expand our understanding of the mechanisms by which the neural circuits responsible for the various modes of sensory perception can cooperate to shape animal behaviour.

## Results

**Food hardness affects food preference**. To examine whether food hardness affects food palatability, we altered the hardness of fly food by changing its agarose content (Supplementary Fig. 1a). Flies given the choice of either 0.5 mM or 1 mM sucrose in food containing the same concentration of agarose prefer the sweeter option (that is, 1 mM sucrose). This is true for both 0.2% (Fig. 1a) and 2% agarose (Fig. 1b). As the hardness of food containing 1 mM sucrose rises, however, the flies shift their preference toward food that is softer but less sweet (Fig. 1a). This is also the case for food containing 0.5 mM sucrose (Fig. 1b). These data suggest flies prefer softer food at the expense of sweetness. It is important to note we found no significant relationship between the amount of food flies ingested and food agarose concentration, suggesting these preferences are not attributable to physical difficulties with swallowing (Supplementary Fig. 1b). Since food hardness affects sweet preference, we next asked whether increased sweetness can overcome the effects of hardness. Given an identical 0.5 mM sucrose concentration, flies prefer softer food (Fig. 1c). As the sucrose level in food containing 2% agarose rises, flies prefer sweeter food despite its hardness (Fig. 1c). This suggests the interaction between food sweetness and hardness is reciprocal.

**Labellar MNs are required for hardness-mediated preference**. Since labellar taste bristles and taste pegs contain MNs, these neurons are likely required for detecting food hardness. To test

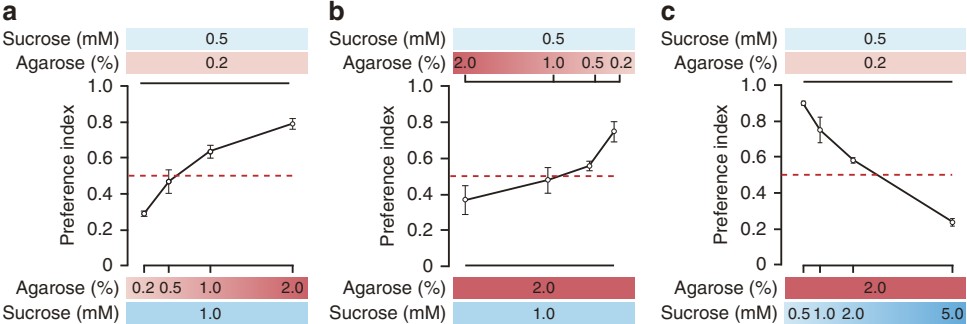

**Figure 1 | Food hardness preference.** (**a,b**) Effect of food hardness on food preference. Flies were given a choice between (**a**) 0.5 mM sucrose in 0.2% agarose and 1 mM sucrose in varying concentrations of agarose (0.2–2%) or (**b**) 0.5 mM sucrose in varying concentrations of agarose (2–0.2%) and 1 mM sucrose in 2% agarose. The red dashed line indicates no preference. n = 4. (**c**) Effect of sweetness on food hardness-based preference. Flies were given a choice between 0.5 mM sucrose in 0.2% agarose and varying concentrations of sucrose (0.5–5 mM) in 2% agarose. n = 4. All data are presented as means ± s.e.m.

this hypothesis, we looked for a mechanosensory *GAL4* line that would permit genetic manipulation of these labellar MNs. After screening several candidate MN *GAL4* lines, we found *VT2692-GAL4* and *R41E11-GAL4* both label a significant proportion of labellar but not tarsal MNs (Fig. 2a). In the labellum, *VT2692-GAL4* and *R41E11-GAL4* are expressed in most labellar taste bristle and taste peg MNs (Supplementary Table 1). We also

found, in addition to its expression in tarsal MNs[9], *R55B01-GAL4* is also expressed in labellar taste bristle MNs (Fig. 2a; Supplementary Table 1).

We confirmed the identity of the *R41E11-GAL4*-expressing cells via cell-type specific marker labelling and morphological analysis. As expected, *R41E11-GAL4* is co-expressed with the labellar MN marker *Fru[LexA]* but not the sweet GRN marker

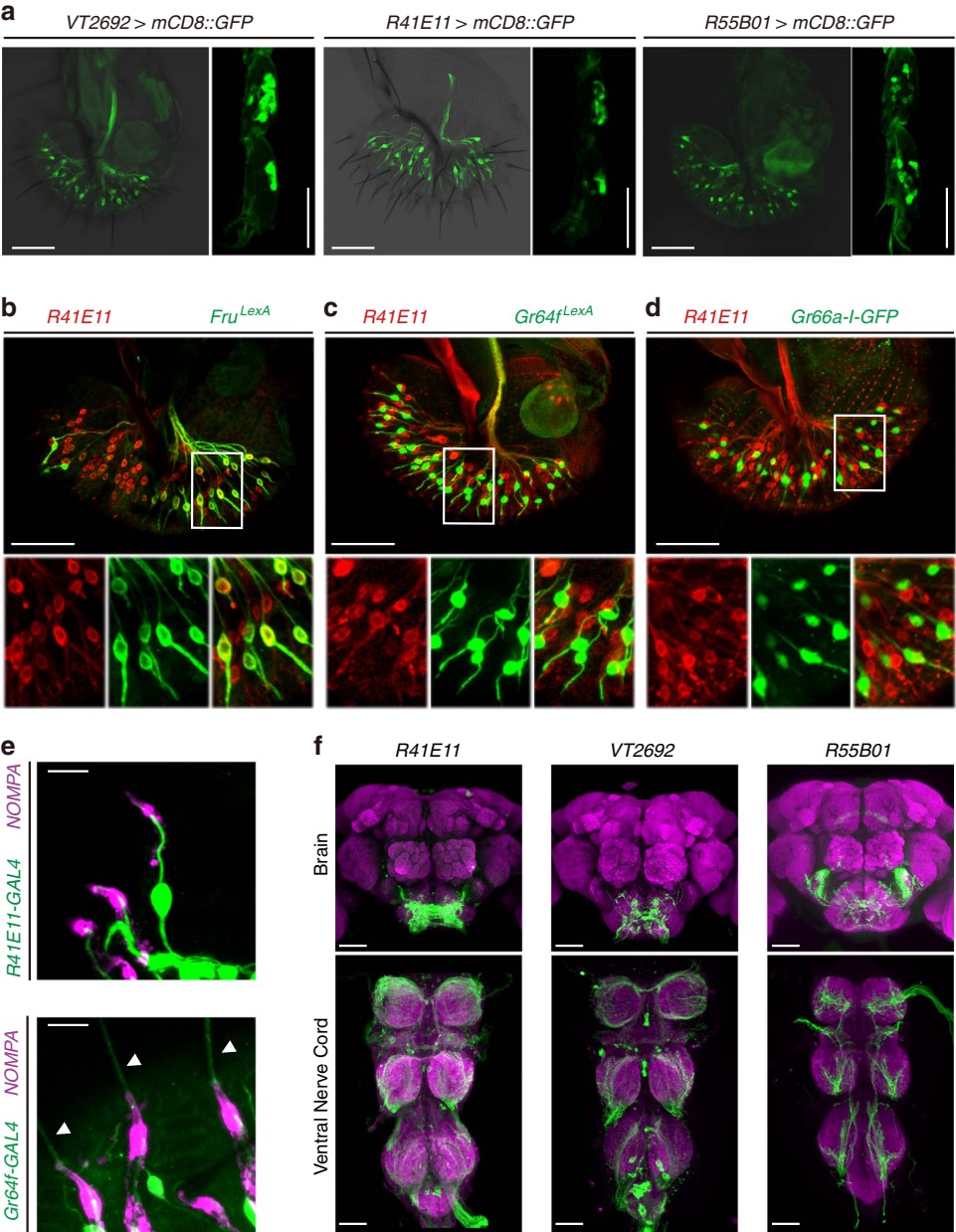

**Figure 2 | Characterization of labellar MN *GAL4* drivers.** (**a**) Expression of MN drivers in the labellum and the leg. Confocal images of labella and legs expressing mCD8::GFP driven by *VT2692-GAL4*, *R41E11-GAL4* and *R55B01-GAL4*. Labella were stained with a rabbit anti-GFP. GFP fluorescence superimposed on a transmitted light image. Scale bars, 50 μm. (**b–d**) Co-localization of R41E11 cells and cell-type specific markers; (**b**) MNs, *Fru[LexA]* (**c**) sweet GRNs, *Gr64f[LexA]* (**d**) bitter GRNs, *Gr66a-I-GFP*. Labellum of *UAS-mCD8::RFP/LexAOp-rCD2::GFP;Fru[LexA]/R41E11-GAL4* flies stained with a rat anti-CD8 and a rabbit anti-GFP. Labellum of *UAS-mCD8::RFP/LexAOp-GCaMP3;Gr64f[LexA]/R41E11-GAL4* and *UAS-mCD8::RFP/ + ;Gr66a-I-GFP/R41E11-GAL4* flies stained with a mouse anti-GFP and a rabbit anti-DsRed. The white boxes indicate the insets shown at higher magnification below. Scale bars, 50 μm. (**e**) Comparison of the dendritic morphology of MNs and sweet GRNs. Labellum of *UAS-DsRed/ + ;NOMPA-GFP/ R41E11-GAL4* (up) or *UAS-DsRed/Gr64f-GAL4;NOMPA-GFP/ +* (bottom) flies stained with rabbit anti-DsRed. The arrowheads indicate the extension of sweet neuron dendrites beyond the bristle base. Scale bars, 10 μm. We used NOMPA as a marker for the bristle base. Note that fluorescent signals from the GAL4 and NOMPA were pseudocoloured green and magenta, respectively, to improve clarity. (**f**) Expression of labellar MN drivers in the central nervous system. Confocal images of brains and ventral nerve cords expressing mCD8::GFP driven by *R41E11-GAL4*, *VT2692-GAL4*, and *R55B01-GAL4*. Samples were stained with rabbit anti-GFP and nc82. Scale bar, 50 μm. All confocal images are maximal intensity projections of *z*-stacks.

$Gr64f^{LexA}$ or the bitter GRN marker $Gr66a$-$I$-$GFP$[10–12] (Fig. 2b–d). In addition, R41E11 cells terminate at the base of bristles labelled with NOMPA, suggesting they are MNs (Fig. 2e). In contrast, $Gr64f^{LexA}$ cells extend their dendrites all the way to the tips of these bristles, confirming their identity as chemosensory neurons (Fig. 2e).

To investigate the role of the labellar MNs in detecting food hardness, we silenced them using the potassium channel Kir2.1 (ref. 13). To rule out any developmental artifacts in this experiment, we used the temperature-sensitive GAL80[ts] to temporally restrict Kir2.1 expression[14]. All flies in this experiment were maintained at 21 °C except for the shift of the experimental group to 31 °C for 3 days before the assay to inactivate GAL80[ts] and silence the MNs via Kir2.1 expression. When given a choice between 0.5 mM sucrose in 0.2% agarose and 1 mM sucrose in 2% agarose at 21 °C, control flies prefer 0.5 mM sucrose in 0.2% agarose. At the restrictive temperature, however, flies with silenced labellar MNs show a significant defect in their preference for the softer food (Fig. 3a). We observed similar results with three independent labellar MN $GAL4$ lines, but not with $Gr68a$-$GAL4$, which labels the tarsal MNs[15] (Fig. 3a). Furthermore, the expression of these three independent $GAL4$ lines in the brain and ventral nerve cord only overlaps in the SEZ (Fig. 2f). These results strongly and specifically implicate the labellar MNs in food hardness detection.

**Nanchung is a mechanosensor detecting food hardness.** To identify the specific mechanosensor required for food hardness detection, we examined the food preference of several mechanosensory mutants. Several mechanosensors have been reported in flies (that is, NompC[16], Nanchung[17], Inactive[18], Pickpocket[19], Painless[20] and Piezo[21]). Only the *nanchung* mutant ($nan^{36a}$) shows diminished preference for softer food (Fig. 3b). To confirm the role of Nan in food hardness detection, we generated another *nan* allele, $nan^{GAL4}$, by homologous recombination (Supplementary Fig. 2a). $nan^{GAL4}$ flies also show reduced preference for softer food (Fig. 3c). This defect is rescued by the expression of a wild-type *nan* cDNA in the mutant background using $R41E11$-$GAL4$ (Fig. 3c). This is strong evidence that Nan is the mechanosensor required in labellar MNs for food hardness detection.

We next used $R41E11$-$GAL4$ to drive the expression of the cell death genes *hid* and *reaper*[22], genetically ablating the labellar MNs. According to a *nan*-specific RT-PCR experiment, labella in which the MNs have been ablated express lower levels of *nan* mRNA than control labella. This confirms the expression of *nan* in the labellar MNs (Supplementary Fig. 2b). We also found $F$-$GAL4$, which expresses GAL4 under the control of the *nan* promoter[17], labels a small subset of labellar MNs (Supplementary Fig. 2c). Silencing these $F$-$GAL4$-expressing labellar MNs, however, does not affect preference for softer food. Since there are so few $F$-$GAL4$-positive labellar MNs, the preference for softer food seems to require activation of a relatively large population of the labellar MNs (Supplementary Fig. 2d).

**MNs inhibit proboscis extension.** When the labellar sensilla of restrained but hungry flies are presented with gustatory stimuli, the flies extend their proboscis in an effort to feed. By quantifying this proboscis extension response (PER), we can measure the desirability of specific gustatory stimuli[23]. To further clarify the nature of the preference for softer food at the expense of sweetness, we next asked whether activation of MNs alters the PER to sweet stimuli. To do so, we used $R41E11$-$GAL4$ to over-express the temperature-sensitive cation channel dTrpA1 ($UAS$-$dTrpA1$)[24] in the labellar MNs. Then, we performed the

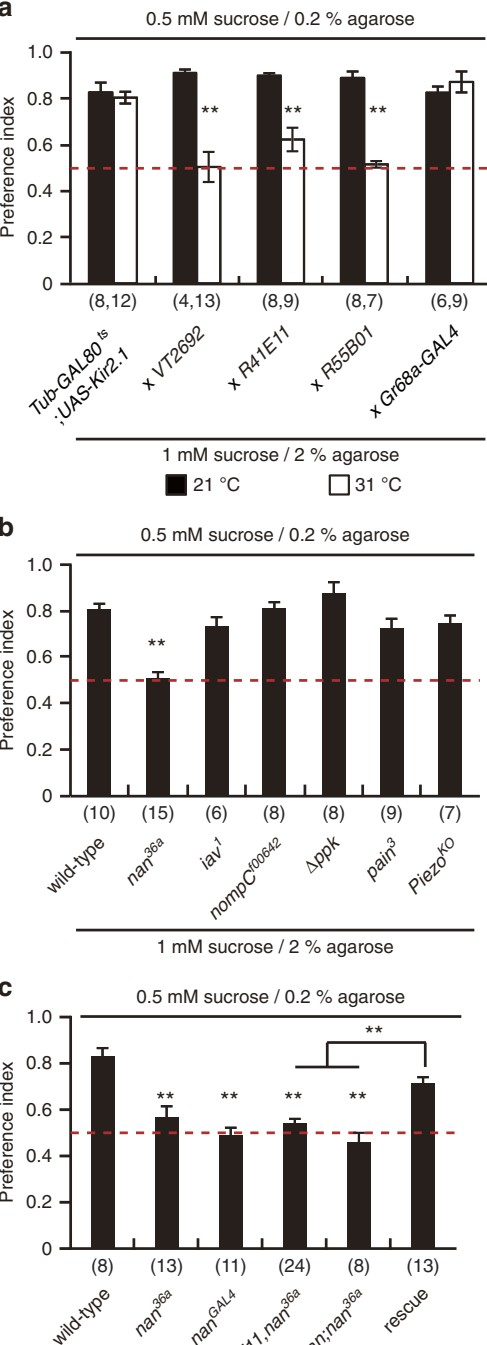

**Figure 3 | Food hardness-mediated preference is regulated by labellar MNs.** (**a**) Food preference on silencing of MNs. Flies were given a choice between 0.5 mM sucrose in 0.2% agarose and 1 mM sucrose in 2% agarose. Flies were raised at 21 °C and shifted to the indicated temperature for 3 days before assaying. Unpaired Student's $t$-tests, $**P < 0.01$. (**b,c**) Mechanosensor screen. Flies were given a choice between 0.5 mM sucrose in 0.2% agarose and 1 mM sucrose in 2% agarose. ANOVA with Tukey *post-hoc* tests, $**P < 0.01$. n is indicated in parentheses. All data are presented as means ± s.e.m.

PER assay at either 21 °C or 31 °C. At 31 °C, dTrpA1 should artificially activate the labellar MNs. As expected, heterozygous control flies (that is, $UAS$-$dTrpA1/+$ and $R41E11$-$GAL4/+$)

show an increase in PER rates with increasing sucrose concentration, which is unaffected by raising the temperature to 31 °C (Fig. 4a). Flies expressing dTrpA1 in their labellar MNs, however, show PER rates to sucrose comparable to those of controls only at 21 °C. At 31 °C, they show significantly reduced

PER rates, suggesting activation of the labellar MNs inhibits sugar sensing (Fig. 4a). We also observed the same effect using the *VT2692-GAL4* and *R55B01-GAL4* labellar MN drivers (Supplementary Fig. 3a). It is noteworthy that neither the activation of the labellar MNs nor the loss of Nan affects

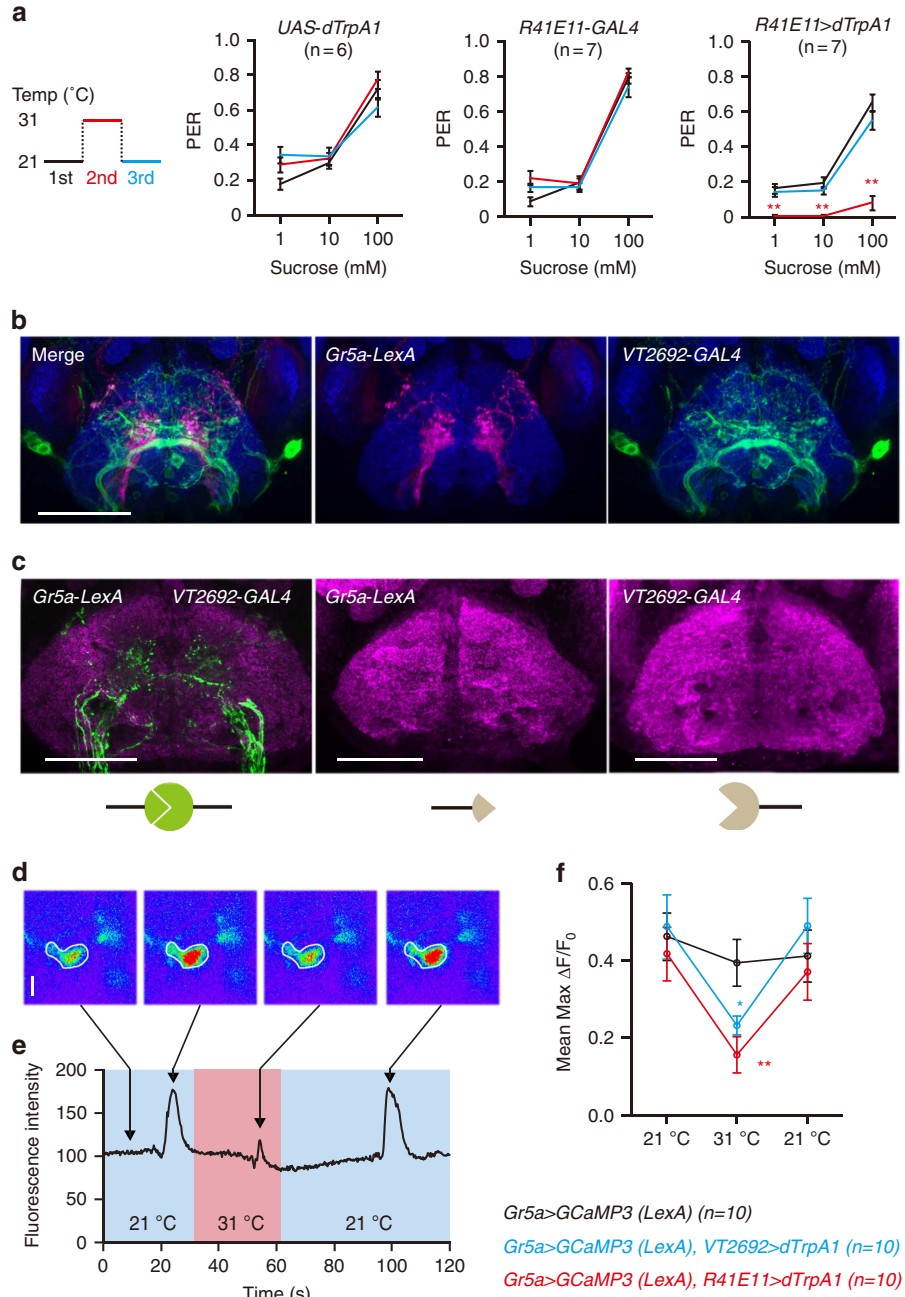

**Figure 4 | Functional interaction between MNs and sweet GRNs in the SEZ.** (**a**) PER on activation of MNs. PER assays were performed at the indicated temperature. ANOVA with Tukey *post-hoc*. **P < 0.01. (**b**) Double-labelling of labellar MN and sweet GRN projections in the SEZ. Brains of *Gr5a-LexA/ + ;UAS-mCD8::GFP/ + ;VT2692-GAL4/LexAOp-mCherryHA* flies were stained with a rabbit anti-GFP (green) and the neuropil marker nc82 (blue). Scale bar, 50 μm. (**c**) GRASP between MNs labelled with *VT2692-GAL4* and sweet GRNs labelled with *Gr5a-LexA*. The genotypes are *Gr5a-LexA/ + ;LexAOp-CD4::spGFP11/ + ;UAS-CD4::spGFP1-10/VT2692-GAL4* (GRASP, left), *Gr5a-LexA/ + ;LexAOp-CD4::spGFP11/ + ;UAS-CD4::spGFP1-10/ +* (LexA only, middle), and *LexAOp-CD4::spGFP11/ + ;UAS-CD4::spGFP1-10/VT2692-GAL4* (GAL4 only, right). Brains were stained with the neuropil marker nc82 (magenta). A schematic for GRASP is shown below. Scale bar, 50 μm. (**d–f**) Measurement of sweet GRN activity in the SEZ on MN activation. GCaMP3 was expressed in *Gr5a-LexA* neurons for calcium imaging and dTrpA1 was expressed in MNs (*VT2692-GAL4* and *R41E11-GAL4*) for artificial neuronal activation. 100 mM sucrose was applied to the labellum. (**d**) Representative pseudocolour images showing sweet GRN activity. Scale bar, 20 μm. SEZ ROIs are outlined in white. Arrows point to the corresponding traces in **e**. (**e**) Representative traces of fluorescence intensity changes evoked by 100 mM sucrose in sweet GRN termini on MN activation. (**f**) Mean maximal fluorescent intensity changes. Each trial was carried out at the indicated temperature. Repeated measure ANOVAs with *post-hoc* Mann–Whitney *U*-tests, *P < 0.05, **P < 0.01. *n* is indicated in parentheses. All data are presented as means ± s.e.m.

feeding as measured by the rate of cibarial pumping[25] (Supplementary Fig. 3b,c).

**Sweet GRN and MN processes are closely apposed in the SEZ.**
GRN axons terminate in the subesophageal zone (SEZ), which is the brain's primary gustatory centre[12,26]. MNs in the taste pegs also send processes to the SEZ[27] suggesting a possible association of chemosensory and mechanosensory signals there. Supporting this, we observed some co-localization in the SEZ after double-labelling the sweet GRNs with Gr5a-LexA[28] and the labellar MNs with VT2692-GAL4 (Fig. 4b). We next performed a GFP Reconstitution Across Synaptic Partners (GRASP)[28] experiment to determine the extent to which the processes of sweet GRNs and labellar MNs are near enough to form synapses. We expressed the two halves of a split GFP in either sweet GRNs using Gr5a-LexA or in labellar MNs using VT2692-GAL4 or R41E11-GAL4. Remarkably, we observed significant GRASP signal in the SEZ even without antibody staining, indicating that these two types of neurons are closely apposed (Fig. 4c; Supplementary Fig. 3d). It is important to note that neither half of the split GFP produced any detectable signal alone, confirming the specificity of the GRASP signal (Fig. 4c).

**MNs inhibit presynaptic calcium responses in sweet GRNs.** We next asked whether the contacts between sweet GRNs and labellar MNs in the SEZ represent functional synapses. To determine this, we measured the calcium responses of sweet GRN axon termini expressing GCaMP3 (ref. 29) under the control of Gr5a-LexA. Simultaneously, we expressed dTrpA1 in labellar MNs using VT2692-GAL4 or R41E11-GAL4 to permit temperature-dependent neuronal activation. Under these conditions, calcium signals in the sweet GRN axon termini should reflect stimulus-driven GRN excitation. We observed that sweet GRN termini show robust calcium responses on stimulation of the proboscis with 100 mM sucrose (Fig. 4d–f). Activation of the labellar MNs by shifting the temperature to 31 °C reduced these calcium signals (Fig. 4d–f; Supplementary Fig. 3e), while a subsequent shift back to 21 °C rescued this reduction. This suggests activation of the labellar MNs inhibits the phagostimulatory effect of sucrose by reducing the presynaptic gain in the sweet GRNs.

**GABA from MNs mediates the inhibition of sweet GRNs.**
GABA, the brain's primary inhibitory neurotransmitter, is known to play a role in presynaptic gain control in sweet GRNs[30]. Since MNs inhibit both the PER to sucrose and the calcium responses of sweet GRN termini, we asked whether GABA is also required for the inhibition of the sweet-evoked appetite response. Indeed, knockdown of vesicular GABA transporter (Vgat) and glutamate decarboxylase1 (Gad1) in the labellar MNs blocks the inhibition of PER induced by activation of MNs (Fig. 5a). This is not the case for labellar MN-specific knockdown of either choline acetyltransferase (Cha) or vesicular acetylcholine transporter (VChaT), both essential for acetylcholine synthesis (Supplementary Fig. 4a). In addition, labellar MN-specific knockdown of either Vgat or Gad1 blunts the preference for softer food at the expense of sweetness (Fig. 5b). We quantified the knockdown efficiency and specificity of the Vgat and Gad1 RNAi lines using quantitative PCR (qPCR) (Supplementary Fig. 4b–d). We were able to confirm the labellar MNs are GABAergic by visualizing their expression of a Vgat-GAL4-driven UAS-mCD8::GFP reporter[31] (Supplementary Fig. 4e). The GFP-labelled neurons showed typical MN morphology (Supplementary Fig. 4f). Finally, we asked which GABA receptor in the sweet GRNs is required for inhibition of sugar sensing. Flies with reduced $GABA_BR_2$ but not $GABA_BR_1$ or $GABA_BR_3$ expression in

sweet GRNs show an impaired shift in the preference for softer food when compared with controls (Fig. 5c). Furthermore, pharmacological inhibition of $GABA_BR_2$ in the SEZ alleviates the suppression of the calcium responses of sweet GRNs on labellar MN activation (Fig. 5d,e). Importantly, knockdown of $GABA_BR_2$ in sweet GRNs does not affect the discrimination of sweeter foods of identical hardness (Supplementary Fig. 4g,h).

**Discussion**
Here, we have uncovered a mechanism by which tactile sensation regulates feeding by controlling the presynaptic gain of phagostimulatory GRNs. Activation of MNs inhibits calcium responses in sweet GRNs via the inhibitory neurotransmitter GABA. This effect likely contributes to Drosophila's preference for ripe or overripe rather than fresh fruits, as both sweetness and hardness change with decay.

The association of MNs with GRNs in labellar taste bristles and taste pegs was first observed several decades ago[8], but the physiologic significance of this association was never investigated. We have shown labellar MNs produce GABA in the SEZ to inhibit signalling through the sweet GRNs. The activation and inhibition of R55B01-GAL4-expressing cells show similar effects on presynaptic gain in sweet GRNs as activation and inhibition of R41E11-GAL4-expressing cells and VT2692-GAL4-expressing cells. This implicates the taste bristle MNs labelled by all three of these lines rather than the taste peg MNs in the interaction between sweet sensing and mechanosensation. The projection of taste peg MNs to an area of the SEZ distinct from that innervated by sweet and bitter GRNs project[27] further supports this idea.

In flies, the tarsal segments of the legs also have chemosensory and mechanosensory sensilla that can be activated during food foraging. Two other groups recently explored the role these tarsal MNs play in behavioural regulation. Ramdya et al.[9] reported that tarsal MNs provide sensory information that drives collective behaviour, and Mann et al.[32] showed that tarsal MNs inhibit feeding via a population of thoracic ganglion interneurons. The fact that the R41E11-GAL4 and VT2692-GAL4 drivers we used in this study are expressed not in the MNs of the legs but in their supporting cells, suggests the tarsal MNs play no role in food hardness detection. In further support of this conclusion, we found inactivation of the tarsal MNs using Gr68a-GAL4 does not impair hardness-mediated food preference (Fig. 2c). Thus, it is clear the tarsal and labellar MNs play different roles in controlling animal behaviour.

Although soft food preference is strongly affected by both silencing of the labellar MNs and the loss of Nan, both of these conditions still show a slight residual preference for soft food (Figs 1 and 3). This suggests the presence of another mechanosensory system involved in food hardness detection, perhaps the pharyngeal MNs[7,33] or labellar multidendritic neurons (Jeong and Moon, unpublished data).

Despite being unable to detect any role for NompC in food hardness detection using our preference assay, NompC's expression in the labellar taste bristle MNs[16,34] makes it a plausible secondary candidate for the labellar MN mechanosensor. In other words, while Nan may act as the mechanosensor in labellar MNs with NompC modulating its function, the reverse may also be true, as is the case in the chordotonal neurons[35,36].

In Drosophila, $GABA_BR_2$ is required in sweet GRN axon termini for the suppression of sweet responses by bitter stimuli when sweet and bitter tastants are mixed together[30]. Knockdown of $GABA_BR_2$ in sweet GRNs increases the PER to sugar as well as to sugar/bitter mixtures. In our study, knockdown of $GABA_BR_2$ in sweet GRNs impairs soft food preference at the expense of sweetness (Fig. 5c), but it does not affect preference for sweetness

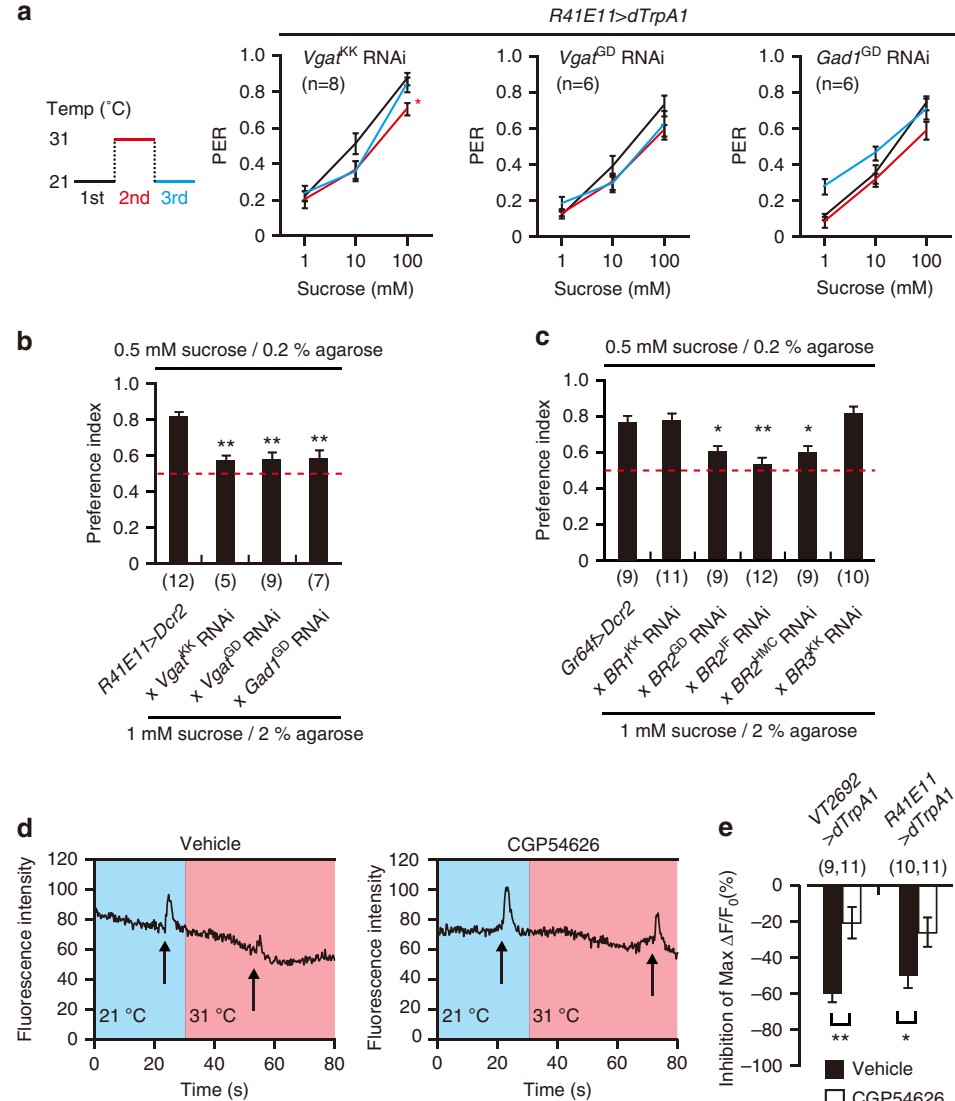

**Figure 5 | GABAergic inhibition of sweet GRNs by MNs.** (**a**) PER in flies with knockdown of GABA-related genes in MNs. Each RNAi line crossed to *UAS-dTrpA1;R41E11-GAL4* was subjected to the PER assay. PER was carried out at the indicated temperature. ANOVAs with Tukey *post-hoc* tests. *$P < 0.05$ (**b,c**) Food preference of flies with (**b**) knockdown of GABA-related genes in MNs and (**c**) knockdown of GABA receptors in sweet GRNs. Flies were given a choice between 0.5 mM sucrose in 0.2% agarose and 1 mM sucrose in 2% agarose. *UAS-Dcr2;R41E11-GAL4* (**b**) and *Gr64f-GAL4;UAS-Dcr2* (**c**) were crossed to each RNAi line. The red dashed line at 0.5 represents no preference. ANOVAs with Tukey *post-hoc* tests. *$P < 0.05$, **$P < 0.01$. (**d,e**) Effect of pharmacological GABA$_B$R blockade in sweet GRN termini on MN-mediated PER inhibition. (**d**) Representative traces for fluorescence intensity changes in sweet GRN termini after stimulation with 100 mM sucrose and MN activation in the presence of the GABA$_B$R antagonist CGP54626. *Gr5a-LexA/ + ;LexAOp-GCaMP3/ + ;UAS-dTrpA1/VT2692-GAL4* were used for calcium imaging. Brains were incubated in artificial haemolymph containing 5 µM CGP54626 for 5 min. (**e**) Normalized inhibition ratio of maximal $\Delta F/F_0$. Unpaired Student's *t*-tests, *$P < 0.05$, **$P < 0.01$. n is indicated in parentheses. All data are presented as means ± s.e.m.

in the absence of differences in food hardness (Supplementary Fig. 4g,h). These data suggest sweet GRNs receive multiple GABAergic inputs from different sensory circuits.

We have shown taste-related mechanosensory information can inhibit sweet perception in the primary taste relay centre, the SEZ, but it remains unclear whether mechanosensation modulates the perception of sweet tastants only at the level of the GRNs or whether the tactile information is relayed to higher brain centres for integration. It will be interesting to see which other parts of the brain these MNs innervate and what other behaviours, apart from food hardness perception, they regulate. It will also be interesting to see whether these or any other MNs interact with taste information in any higher brain centres. During feeding, multiple modes of sensory information must be perceived and

integrated to produce the perception of 'flavour'. This phenomenon is well-described in humans using mainly a psychophysiological approach, but the molecular mechanisms and neural circuits that produce it remain unclear. Using the *Drosophila* model system, we have explored potential circuit motifs underlying multimodal sensory processing and have demonstrated an intriguing interaction between sweet GRNs and MNs that modulates feeding decision-making.

## Methods

**Fly stocks.** All fly stocks were raised in standard cornmeal-molasses-agar medium and maintained with a 12 h/12 h light/dark cycle at 25 °C and 60% humidity. We obtained the following lines from the Bloomington stock centre: *70FLP,70I-SceI/CyO, Df(3L)BSC601, Tub-GAL80$^{ts}$, UAS-Kir2.1, UAS-hid, UAS-reaper,*

Gr64f-GAL4, Gr68a-GAL4, F-GAL4, nan[36a], iav[1], nompC[f00642], Df(2L)A400T (2:3:4), Df(2L)b88h49, pain[3], Piezo[KO], UAS-nan, UAS-Dicer2, UAS-mCD8::GFP, UAS-mCD8::RFP, UAS-DsRed, NOMPA-GFP, LexAOp2-rCD2::GFP, LexAOp2-mCherryHA, LexAOp-GCaMP3, UAS-dTrpA1, R41E11-GAL4, R55B01-GAL4, Vgat-GAL4 (BL58980), GABA$_B$R2[HMC] (BL50608), GABA$_B$R2[JF] (BL27699), Cha[JF] (BL25856), and VAcht[JF] (BL27684). We obtained the following lines from the VDRC stock centre: Vgat[KK] (v103586), Vgat[GD] (v45916), Gad1[GD] (v32344), GABA$_B$R1[KK] (v101440), GABA$_B$R2[GD] (v1785), GABA$_B$R3[KK] (v108036), and VT2692-GAL4 (v205409). Gr66a-I-GFP, Gr5a-LexA, LexAOp-CD4::spGFP$_{11}$, and UAS-CD4::spGFP$_{1-10}$ were a gift from K. Scott. Gr64f[LexA] and Fru[LexA] were from H. Amrein and B. Baker, respectively.

**Genetics.** nan[GAL4] was generated by ends-out homologous recombination[37]. Overall, 3 kb 5′- and 3′-end homology arms were obtained by PCR amplification from isogenized w[1118] genomic DNA with the following primers: (5′arm: 5′-GGAT CCGGGGAATGCGAAAATCAACAAATAAAT-3′ and 5′-GGTACCCATTAT CCGATCCCAAATTCACTC-3′; 3′ arm: 5′-GCGGCCGCGTTGGAACAAACA TACACATAAAAAAC-3′ and 5′-CCGCGGTGGCAAAGCTGTTTAATTAC GCCCC-3′). Each arm was then inserted into the pw35GAL4 vector. The final targeting construct was inserted by germ line transformation into the w[1118] background (Bestgene, Inc., Chino Hills, CA, USA). Although we placed the GAL4 start codon in-frame with that of the nan sequence, we did not observe any GAL4 expression. Transgenic flies were crossed to 70FLP,70I-SceI/CyO to excise the targeting construct from the genome of the transgenic flies. Mosaic-eyed progeny (F1) were crossed to w[1118] to obtain red-eyed F2 progeny. Selected F2 progeny were crossed to Df(3L)BSC601 and their progeny were subjected to PCR analysis with the following primers: 5′-GAGGCCGAGTATATCTCCAATCC-3′ and 5′-CTCGTAGCCAACATCGAACATTTCGATC-3′.

**RT-PCR.** Total RNA was extracted from 300 dissected labella per each genotype using TRIZOL (Invitrogen, Carlsbad, CA, USA). A total volume of 2.5 μg of this total RNA was used for cDNA synthesis using RevertAid reverse transcriptase (Thermo Fisher Scientific). The following primers were used to detect nan and rp49 transcripts: (nan: 5′-GAGGCCGAGTATATCTCCAATCC-3′ and 5′-AGCAG GCACAAATGGAGAATAGTT-3′), (rp49: 5′-GACCATCCGCCCAGCATACA G-3′ and 5′-AATCTCCTTGCGCTTCTTGGAGGAG-3′).

**Quantitative PCR.** qPCR was performed with an ABI7500 real-time PCR machine (Applied Biosystems) using the ABI SyBr Green system. The $C_T$ (threshold cycle) for each transcript was averaged from technical duplicates from three independent biologic samples and normalized to the $C_T$ for the rp49 internal control. We used the ΔΔCT method for comparing relative expression. We used the following primers to detect Gad1 and Vgat transcripts: (Gad1: 5′- CAAGTTAAGACGGG ACATCCCCACTTC-3′ and 5′- GCATCTTGGTCAGCACCACATTCTC-3′), (Vgat: 5′-GACGGCTTTAGGCAAGGTAGCATC -3′ and 5′-GAACATGCCC TGAATGGCATTGGTC -3′). We used the rp49 and nan primers used for RT-PCR.

**Chemicals.** Sucrose and sulforhodamine B were purchased from Sigma-Aldrich (Saint Louis, MO, USA). Brilliant Blue FCF was purchased from Wako Pure Chemical Industries, Ltd (Osaka, Japan). Agarose was purchased from Invitrogen (Cat no. 75510-019, Carlsbad, CA, USA). CGP54626 was purchased from Tocris (Bristol, UK).

**Two-way choice assay.** To form the two-choice plates, we poured one type of food into 60 mm$^2$ dishes and allowed it to solidify before cutting half away and filling the empty space with another type of food. Flies were subjected to the feeding assay after both types of food were completely solidified. Overall, 3–5-day-old flies starved for 18 h were allowed to choose between blue or red food in a dark room for 90 min. After feeding, we observed the colour of frozen fly abdomens by light microscopy and calculated a preference index (P.I.) using the following equation: P.I. = [(# of red abdomens) + 0.5 × (# of purple abdomens)]/(# of coloured abdomens).

**PER.** Overall, 3–5-day-old flies starved for 18 h were glued to a slide glass after ice anaesthesia in groups of 24 per genotype. All flies were sated with water before assaying. A 1 ml syringe was used to apply a droplet of sucrose solution to the labellum. To activate the dTrpA1-expresssing neurons, the slide glass with flies attached was heated on a heat block (31 °C), while the air temperature was maintained at 21 °C to minimize any artifactual stimulation.

**Measurement of feeding amount.** Each concentration of sucrose was dissolved into an indicated concentration of agarose gel in fly vials. The food was presented to 18 h-starved flies for 90 min in a dark room. Frozen fly bodies were vortexed to remove the heads after brief exposure to liquid nitrogen. The bodies were then collected and ground in PBS (10 μl PBS per fly body). OD630 of a final 100 μl

solution was measured in 96-well plates. A basal OD630 of an empty well was subtracted from each OD630 result.

**Pumping assay.** The flies used in the pumping assay were prepared in a way similar to those used in the PER assay. A total of 500 mM sucrose solution with Brilliant Blue FCF (5 mg ml$^{-1}$) was applied to the labellum using a 1 ml syringe. To activate the dTrpA1-expresssing neurons, the slide glass with flies attached was heated on a heat block (31 °C). The heat block was maintained at 21 °C for the controls. Fly feeding behaviour was recorded at 24 fps using a SONY HDR-XR520 Handycam through a NIKON SMZ645 stereomicroscope. Pixel intensity changes were measured in an appropriate ROI around the cibarium using ImageJ. Pumping frequency was calculated based on the initial 5 s of pumping behaviour.

**Immunostaining.** Dissected tissues were fixed in 4% paraformaldehyde (PF) in 0.2% Triton X-100 PB (PBT) for 20 min following three washes in 0.2% Triton X-100 PBS (PBST) for 10 min each. Samples were blocked in 5% goat serum PBST for 20 min. Primary antibodies were incubated in PBST overnight at 4 °C. Titres for primary antibodies were as follows: mouse nc82 (1:50, DSHB), mouse anti-GFP (1:200, A11120, Invitrogen) and rabbit anti-GFP (1:500, A11122, Invitrogen), rat anti-CD8 (1:200, MCD800, CALTAG), rabbit anti-NOMPA[38] (1:200) and rabbit anti-DsRed (1:200, 632496, Clontech). Secondary antibodies were incubated in PBST for 1 h at room temperature after three washes. Secondary antibodies were Alexa 488 anti-rabbit, Alexa 488 anti-mouse, Alexa 568 anti-mouse, Alexa 568 anti-rabbit and Alexa 647 anti-rat (1:400, A11034, A11029, A11031, A11011, A21247, Molecular Probes). Samples were mounted with Vectashield (Vector Laboratories, Burlingame, CA, USA) and observed under a LSM 700 Zeiss confocal microscope (Jena, Germany).

**GRASP.** Brains were fixed in 4% PF/0.2% Triton X-100 PB and washed three times for 10 min each. To minimize false positive signals, we did not stain with an anti-GFP antibody. The neuropil was counter-stained with nc82 to visualize gross brain morphology. Samples were mounted with Vectashield (Vector Laboratories, Burlingame, CA, USA) and observed under a LSM 700 Zeiss confocal microscope (Jena, Germany).

***In vivo* calcium imaging.** Overall, 2-week-old flies were mounted on a customized imaging chamber after brief ice anaesthesia. All legs were removed using scissors. The cuticle and connective tissue covering the SEZ was removed using fine forceps. The proboscis was extended with forceps and immobilized in a collar of wax. The brain and chamber were filled with artificial haemolymph (AHL: 108 mM NaCl, 5 mM KCl, 2 mM CaCl$_2$, 8.2 mM MgCl$_2$, 4 mM NaHCO$_3$, 1 mM NaH$_2$PO$_4$, 15 mM ribose, 5 mM HEPES). A coverslip was placed obliquely to isolate the proboscis from the buffer-filled space. The AHL-exposed brain was covered with another coverslip for visualization under a confocal microscope. GCaMP3 fluorescence was imaged by a Plan-Apochromat 10 × /0.45 M27 lens. The ROI in the SEZ was digitally magnified three to four times, and visualized at a resolution of 256 × 256 pixels, one frame every 0.243 s . The pinhole was opened to 150 μm. GCaMP3 fluorescence was imaged from 10 s before application of taste solution to at least 10 s after peak fluorescence intensity. A droplet of taste solution was delivered by a 1 ml syringe for about 1 s . For quantification, $\Delta F/F_0$ was calculated using the following equation: $\Delta F/F_0 = ((\text{single frame intensity}) - (\text{average intensity of 10 frames just before tastant application}))/(\text{average intensity of 10 frames just before tastant application})$. For our pharmacological approach, the fly brain mounted for imaging was pre-incubated with either 5 μM CGP54626 or vehicle (DMSO) for 5 min. Room temperature was maintained at 21 °C. To activate dTrpA1, warm air was applied to the fly head using an air heater and the temperature near the fly was monitored by an electronic thermometer.

**Measurement of food hardness.** Agarose gel was prepared from a 6 ml solution in a 60 mm$^2$ diameter dish (Nunc, Denmark). Agarose hardness was measured with a TA.HD Plus Texture Analyser (Stable Micro Systems, UK) using a 20 mm aluminium cylinder probe with the following instrument settings: measured force in compression, pre-test speed: 1.0 mm s$^{-1}$; test speed: 2.0 mm s$^{-1}$; post-test speed: 10.0 mm s$^{-1}$; strain; 50%; trigger force: 10g. The maximum force correlates to the hardness of the sample. All textural analyses were carried out using Texture Exponent software version 6.1.5.0. (Stable Micro Systems). The sample was positioned centrally under the probe during testing.

**Statistics.** All error bars indicate s.e.m. Normality and homoscedasticity were tested using the Kolmogorov–Smirnov test and Levene's test, respectively. We used ANOVAs with Tukey *post-hoc* tests to analyze most of the two-way choice assays, pumping assays and PER results. The unpaired Student's *t*-test was used for Fig. 3a; Supplementary Figs 2d and 3b,c. Repeated measure ANOVAs with *post-hoc* Mann–Whitney *U*-tests and unpaired Student's *t*-tests were used for analyzing calcium imaging data, in Figs 4f and 5e, respectively. qPCR data were analyzed by paired Student's *t*-test or ANOVAs with Tukey *post-hoc* tests. Asterisks indicate *$P < 0.05$ and **$P < 0.01$.

**Data availability.** All relevant data are available from the authors upon request.

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

## Acknowledgements

We thank Drs K. Scott, H. Amrein, B. Baker, the Bloomington Stock Center and the Vienna *Drosophila* RNAi Center for providing fly stocks. We also thank Dr Y.D. Chung for sharing NOMPA antibodies. This work was supported by the National Research Foundation of Korea (NRF) Grant funded by the Korean Government (MSIP) (NRF-2014R1A2A1A11050045 and NRF-2016R1A5A2008630).

## Author contributions

Y.T.J. designed and conducted most of the experiments. S.M.O. did some of the immunostaining experiments. J.S. and Singh, R.N. measured food hardness. J.T.S., J.Y.K. and S.J.M. supervised the project and wrote the paper.

## Additional information

**Competing financial interests:** The authors declare no competing financial interests.

