## [Peer Review File · Nature Communications]

Redactions:

When text has been deleted in rebuttals and referee reports, "[redacted]" has been added in that location.

Reviewers' comments:

Reviewer #1 (Remarks to the Author):

It has been known that all the taste chemosensilla of flies are associated with one mechanosensory neuron, but their real function has never been demonstrated. In this manuscript Jeong and colleagues clearly reveal the function of the mechanosensory neuron in the taste system of *Drosophila*. First they show that sweetness is weakened by hardness of food by using a behavioral feeding assay. Then they screened Gal4 lines whose expression is in the labellar mechanosensory neurons and by using them they could silence the neuron and confirms the inhibitory action of mechano-information on the sweetness pathway. Next problem is to identify the mechanosensitive receptor and they successfully show that Nanchung is the receptor and confirm the conclusion by a rescue experiment. They also show that activation of these neurons inhibit sweet sensation. Finally they demonstrate that the mechanosensitive neurons directly contact with sweet-sensitive neurons and GABA is the transmitter acting from the mechanosensitive neuron. I think every piece of experimental results is obtained by carefully designed experiments and results are convincing. I suggest several points that should be considered for revision of this manuscript.

1. Stimulation of labellar chemosensilla induces opening of the labellar lobes, thus the mechanosensory inhibition will only work on this process. If the mechanosensory inhibition occurs in the taste peg neurons, sucking of sugar-agar will be inhibited. The authors suggest that the mechanosensory neurons in labellar chemosensilla, but not taste peg mechanosensory neurons are involved in the modulation based on the fact that the taste peg mechanosensory neurons project to the SEG region distinct from the labellar neurons projection region. However, there are sweet-sensing neurons on the taste peg projecting to the same region as the taste peg mechanosensory neurons (I am not sure whether this is already known). Also the mechanosensory neurons are also present in pharyngeal taste organs. The authors show data on the amount of intake of different agar concentrations of sugar, but this result does not exclude the possibility that mechanosensation is involved in sucking. They should discuss about these aspects.
2. The authors discuss sensory integration between insects and mammals. The taste information processing in insects is quite different from that in mammals since integration takes place more centrally and I think it is not easy to compare the multi-sensory integration between insects and mammals.
3. The authors find that Nanchung is functioning in the labellar mechanosensory neuron. In other mechanosensilla *nompC* is reported to be expressed. They might discuss how and why mechanosensory neurons are heterogeneous and if the labellar mechanosensation has any unique property. On this regard, it is interesting to know where Nanchung is expressed.

4. The result in Figure 1a shows that the preference value between 0.5 mM suc/0.2% agar and 1 mM suc/0.2 % agar is around 0.3. This PI value would be expected if flies have no mechanosensation. However, PI values in the later experiments on Nanchung-mutants and Kir suppression experiments PIs are around 0.5. If these differences are significant, these results might suggest the presence of an additional mechanosensing pathway.

minor points;

Figure 1: It is strange to abbreviate agar as "agr".

Figure 2 legend (a): change rabbit GFP to rabbit anti-GFP

There are several reference lists (5, 6) to be reformatted.

.

Reviewer #2 (Remarks to the Author):

This manuscript proposes to study the role of mechanosensory neurons in sweet taste detection in *Drosophila*. The authors show that flies prefer softer food to harder food. They identify three Gal4 lines expressed in labellar neurons and show that silencing them causes loss of preference to softer food. Nanchung mutants lacking a mechanosensory channel show the same phenotype. Activating these neurons inhibits the response to sucrose behaviorally and by calcium imaging, and the authors provide evidence that this is mediated by GABA signaling. The authors propose that mechanosensory neurons feed back onto sweet gustatory axons to inhibit the sugar response. The notion that mechanosensory information influences taste perception is exciting. However, there are significant limitations in the experiments and the data is too preliminary for the conclusions drawn.

1. The three Gal4 lines used in this study do not appear to selectively label mechanosensory neurons. Double labeling with bitter, sugar, water, and mechanosensory markers would be required to make this evaluation. Moreover, the mechanosensory neurons should terminate at the base of the bristle and express Nanchung. Do VT26T2, R41E11, and R55B01 all label the same neurons? High-resolution images showing that all three lines are specifically expressed in mechanosensory neurons is necessary to evaluate this study. As it is, it is unclear how many different labellar subtypes are labeled in these Gal4 lines. Fig 2B suggests that R41E11 is expressed in more than mechanosensory neurons, but there is not enough evidence to evaluate this either way.

2. A complete description of each Gal4 line showing expression in the brain and ventral nerve cord is required. If these lines do not selectively and exclusively label mechanosensory neurons, how can conclusions about causal neurons be made?

3. Fig 3 D-F should be performed with P2X2 or Chrimson activation. Heat application seems problematic, what is the evidence that this selectively activates the TRPA1 neurons and not other neurons? Does heat cause calcium responses in T26T2, R41E11, and R55B01 neurons?

4. The activation and RNAi experiments are not well-controlled. Are the labellar neurons in VT26T2, R41E11, and R55B01 GABAergic by immunostaining? Does the RNAi knock down expression of targeted genes? Does expression of RNAi affect expression of the TRP channel?

Reviewer #1:

It has been known that all the taste chemosensilla of flies are associated with one mechanosensory neuron, but their real function has never been demonstrated. In this manuscript Jeong and colleagues clearly reveal the function of the mechanosensory neuron in the taste system of *Drosophila*. First they show that sweetness is weakened by hardness of food by using a behavioral feeding assay. Then they screened Gal4 lines whose expression is in the labellar mechanosensory neurons and by using them they could silence the neuron and confirms the inhibitory action of mechano-information on the sweetness pathway. Next problem is to identify the mechanosensitive receptor and they successfully show that Nanchung is the receptor and confirm the conclusion by a rescue experiment. They also show that activation of these neurons inhibit sweet sensation. Finally they demonstrate that the mechanosensitive neurons directly contact with sweet-sensitive neurons and GABA is the transmitter acting from the mechanosensitive neuron. I think every piece of experimental results is obtained by carefully designed experiments and results are convincing. I suggest several points that should be considered for revision of this manuscript. Major points;

1. Stimulation of labellar chemosensilla induces opening of the labellar lobes, thus the mechanosensory inhibition will only work on this process. If the mechanosensory inhibition occurs in the taste peg neurons, sucking of sugar-agar will be inhibited. The authors suggest that the mechanosensory neurons in labellar chemosensilla, but not taste peg mechanosensory neurons are involved in the modulation based on the fact that the taste peg mechanosensory neurons project to the SEG region distinct from the labellar neurons projection region. However, there are sweet-sensing neurons on the taste peg projecting to the same region as the taste peg mechanosensory neurons (I am not sure whether this is already known). Also the mechanosensory neurons are also present in pharyngeal taste organs. The authors show data on the amount of intake of different agar concentrations of sugar, but this result does not exclude the possibility that mechanosensation is involved in sucking. They should discuss about these aspects.

To address this issue, we examined the effect of the labellar mechanosensory neurons on cibarial pumping. Since neither activation of the mechanosensory neurons labelled by R41E11 or VT2692 nor the loss of Nan affects cibarial pumping, it is unlikely that the labellar mechanosensory neurons are involved in other aspects of feeding behavior. These data are now included in the manuscript in Supplementary Fig. 3b,c and described on page 9 as follows:

“It is noteworthy that neither the activation of the labellar mechanosensory neurons nor the loss of Nan affects feeding as measured by the rate of cibarial pumping (Supplementary Fig. 3b,c).”

We also wanted to explain more clearly why we suggest the labellar mechanosensory neurons are the main player in controlling the presynaptic

gain of the sweet GRNs. To this end, we have newly added Supplementary Table 1 and modified the discussion as follows:

“The activation and inhibition of R55B01-GAL4-expressing cells show similar effects on presynaptic gain in sweet GRNs as activation and inhibition of R41E11-GAL4-expressing cells and VT2692-GAL4-expressing cells. This implicates the taste bristle mechanosensory neurons labeled by all three of these lines rather than the taste peg mechanosensory neurons in the interaction between sweet sensing and mechanosensation. The projection of taste peg mechanosensory neurons to an area of the SEZ distinct from that innervated by sweet and bitter GRNs project27 further supports this idea.”

2. The authors discuss sensory integration between insects and mammals. The taste information processing in insects is quite different from that in mammals since integration takes place more centrally and I think it is not easy to compare the multi-sensory integration between insects and mammals.

We have removed the discussion of mammalian systems from the Discussion section.

3. The authors find that Nanchan is functioning in the labellar mechanosensory neuron. In other mechanosensilla *nompC* is reported to be expressed. They might discuss how and why mechanosensory neurons are heterogeneous and if the labellar mechanosensation has any unique property. On this regard, it is interesting to know where Nanchan is expressed.

Nanchung GAL4 (F-GAL4) is expressed in a subset of labellar mechanosensory neurons, but this may not faithfully represent the true *nan* expression pattern because silencing of the F-GAL4 cells has no effect on food hardness-mediated preference. We now show that genetic ablation of the labellar mechanosensory neurons using R41E11-GAL4 (R41E11-GAL4/UAS-*hid*,UAS-*reaper*) reduces *nan* expression. Furthermore, introduction of a wild-type *nan* to the *nan* mutant background using the labellar mechanosensory neuron drivers rescues the mutant phenotype. These data strongly suggest that *nan* is expressed in the labellar mechanosensory neurons. Unfortunately, we were unable to visualize Nan's localization because no Nan-specific antibody is available. We have added these results to Supplementary Fig. 2b-d and described them on pages 8–9.

Since NOMPC is reportedly expressed in labellar mechanosensory neurons, NOMPC and Nan likely function in the same cells. The role of these two different mechanoTRP channels may be completely distinct or they may function together. Although we only found evidence of Nan's involvement in food hardness detection, it is possible that Nan serves as the primary sensor and NOMPC as a modifier or vice versa. We have now added the following to the Discussion section:

“Despite being unable to detect any role for NompC in food hardness detection using our preference assay, NompC’s expression in the labellar taste bristle mechanosensory neurons makes it a plausible secondary candidate for the labellar mechanosensory neuron mechanosensor. In other words, while Nan may act as the mechanosensor in labellar mechanosensory neurons with NompC modulating its function, the reverse may also be true, as is the case in the chordotonal neurons.”

4. The result in Figure 1a shows that the preference value between 0.5 mM suc/0.2% agar and 1 mM suc/0.2 % agar is around 0.3. This PI value would be expected if flies have no mechanosensation. However, PI values in the later experiments on Nanchan-mutants and Kir suppression experiments PIs are around 0.5. If these differences are significant, these results might suggest the presence of an additional mechanosensing pathway.

We agree and have added the following sentences to page 14 in the Discussion section:

“Although soft food preference is strongly affected by both silencing of the labellar mechanosensory neurons and the loss of Nan, both of these conditions still show a slight residual preference for soft food (Figs. 1 and 3). This suggests the presence of another mechanosensory system involved in food hardness detection, perhaps the pharyngeal mechanosensory neurons or labellar multidendritic neurons.”

Minor points;

Figure 1: It is strange to abbreviate agar as "agr".

We have corrected this.

Figure 2 legend (a): change rabbit GFP to rabbit anti-GFP

We have corrected this.

There are several reference lists (5, 6) to be reformatted.

We have reformatted the references.

Reviewer #2:

This manuscript proposes to study the role of mechanosensory neurons in sweet taste detection in *Drosophila*. The authors show that flies prefer softer food to harder food. They identify three Gal4 lines expressed in labellar neurons and show that silencing them causes loss of preference to softer food. Nanchung mutants lacking a mechanosensory channel show the same phenotype. Activating these neurons inhibits the response to sucrose behaviorally and by calcium imaging, and the authors provide evidence that this is mediated by GABA signaling. The authors propose that mechanosensory neurons feed back onto sweet gustatory axons to

inhibit the sugar response. The notion that mechanosensory information influences taste perception is exciting. However, there are significant limitations in the experiments and the data is too preliminary for the conclusions drawn.

1. The three Gal4 lines used in this study do not appear to selectively label mechanosensory neurons. Double labeling with bitter, sugar, water, and mechanosensory markers would be required to make this evaluation. Moreover, the mechanosensory neurons should terminate at the base of the bristle and express Nanchung. Do VT26T2, R41E11, and R55B01 all label the same neurons? High-resolution images showing that all three lines are specifically expressed in mechanosensory neurons is necessary to evaluate this study. As it is, it is unclear how many different labellar subtypes are labeled in these Gal4 lines. Fig 2B suggests that R41E11 is expressed in more than mechanosensory neurons, but there is not enough evidence to evaluate this either way.

As suggested, we have performed double-labeling experiments with bitter GRN, sweet GRN, and mechanosensory neuron markers (Fru^{LexA}). Please see the new Fig. 2b-d. R41E11-GAL4 cells only overlap with Fru^{LexA} -positive cells. Reviewer #2 is concerned that our mechanosensory neuron drivers label may label several cell types because Fru^{LexA} is only expressed in the dorsal taste bristle mechanosensory neurons. We have changed Fig. 2b to include the whole labellum. This will make it easier for readers to evaluate the expression pattern of each reporter. In addition, we have performed the suggested morphological analysis. R41E11-GAL4-positive neurons and mechanosensory neurons are morphologically distinct from chemosensory neurons. We have added these data to Fig. 2e and described them on page 7 as follows:

“In addition, R41E11 cells terminate at the base of bristles labeled with NOMPA, suggesting they are mechanosensory neurons. In contrast, $Gr64^{LexA}$ cells extend their dendrites all the way to the tips of these bristles, confirming their identity as chemosensory neurons (Fig. 2e).”

2. A complete description of each Gal4 line showing expression in the brain and ventral nerve cord is required. If these lines do not selectively and exclusively label mechanosensory neurons, how can conclusions about causal neurons be made?

We have added the expression of each GAL4 line in the brain and ventral nerve cord as requested. There are very few labeled cells in the central brain, suggesting our phenotype is not due to some non-specific effect of silencing unknown cells. These data are now included in Fig. 2f and described on page 8.

3. Fig 3 D-F should be performed with P2X2 or Chrimson activation. Heat application seems problematic, what is the evidence that this selectively activates the TRPA1 neurons and not other neurons? Does heat cause calcium responses in T26T2, R41E11, and R55B01 neurons?

We found that heat application in the absence of dTrpA1 expression does not induce any calcium response in GCaMP3-expressing sweet GRNs. Ni et al (Nature 2013) and Kang et al (Nature 2011) also show labellar chemosensory neurons are not activated by increases in temperature of up to 31 °C. We also examined the calcium responses of mechanosensory neurons upon heat application. Only mechanosensory neurons expressing dTrpA1 show calcium responses upon heat application (See below).

4. The activation and RNAi experiments are not well-controlled. Are the labellar neurons in VT26T2, R41E11, and R55B01 GABAergic by immunostaining? Does the RNAi knock down expression of targeted genes? Does expression of RNAi affect expression of the TRP channel?

Unfortunately, we were unable to observe GABA staining in the labellar mechanosensory neurons. The RNAi lines we used to knockdown GABA-related genes, however, have already been confirmed by many researchers. It is unlikely knockdown of either *Vgat* or *Gad1* with multiple RNAi lines targeting different portions of the same gene (e.g., *VgatKK* and *VgatGD*) have the same off-target effects. It is even more unlikely that they would also show the same effects as *nan* loss-of-function. In addition, the knockdown of cholinergic components has no effect on the suppression of sugar sensing by mechanosensory neurons. This confirms that our observations are not some sort of non-specific effect of RNAi. Furthermore, because pharmacological blockade of the GABAB receptor abolishes the inhibition of sweet sensing by mechanosensory neurons, we are confident in our conclusion that the mechanosensory neurons are GABAergic.

Reviewers' comments:

Reviewer #1 (Remarks to the Author):

The authors performed additional experiments to reply to comments raised by referees. Most of results added seem to be satisfactory in supporting their conclusion, however, the authors should consider and revise the following points.

Comment#1

The authors assume that cibarial pumping speed is dependent on the sweetness of sugar solution. This assumption must be supported by experimental evidence. Also the "taste peg neurons project SEZ region distinct from labellar neurons" should be followed by a reference.

Comment #3

A reference is needed for "NOMPC is reportedly expressed in labellar mechanosensory neurons" and for "the reverse may also be true, as is the case in the chordotonal neurons". The authors suggest that NOMPC might function with NAN, while *nompC* mutant flies show normal discrimination of agar softness. How can this be explained?

Page 8, line 8,

PIEZO should be "Piezo".

Comment #4

The authors argue that "perhaps the pharyngeal mechanosensory neurons or labellar multidendritic neurons." are also involved. Presence of both kinds of neurons should be referred. In addition mechanosensation in these neuron might not be NAN-dependent.

Table S1.

More explanation is needed. The numbers shown here must be no. of GFP-positive mechanosensory cells and what is the total number of mechanosensory cells in the labellum and in the taste peg? It should be important to know what percentage of cells is GFP-positive.

Reviewer #2 (Remarks to the Author):

The revised manuscript has improved but failed to address one critical concern. The experiments suggesting that the mechanosensory neurons are GABAergic and inhibit sugar neurons are not well-executed and none of the suggested controls were done.

1) There is no evidence that any mechanosensory neuron in *Drosophila* is GABAergic. This would need to be demonstrated by immunostaining. (GAD-Gal4 may also be acceptable)

2) There is a lot of evidence that GABA feeds back onto sugar cells to inhibit their activity through GABA interneurons. It seems plausible that a similar mechanism operates here. As the R41E11 line is expressed in many neurons in the SEZ, it is possible that GABA RNAi is impacting interneurons rather than mechanosensory neurons.

My strong recommendation would be either

A) that the authors show that the mechanosensory neurons are GABAergic and enzymes are knocked down in mechanosensory neurons

B) or that they eliminate Fig 5A and B until they can demonstrate the existence of GABA mechanosensory neurons.

Point-by-point response

Reviewer #1:

The authors performed additional experiments to reply to comments raised by referees. Most of results added seem to be satisfactory in supporting their conclusion, however, the authors should consider and revise the following points.

Comment#1

The authors assume that cibarial pumping speed is dependent on the sweetness of sugar solution. This assumption must be supported by experimental evidence. Also the "taste peg neurons project SEZ region distinct from labellar neurons" should be followed by a reference.

We showed that neither activation of the labellar mechanosensory neurons nor loss of *nan* affect cibarial pumping. We did this to rule out the possibility that the food hardness-dependent preference we see is due to a problem with food ingestion. Manzo et al. (PNAS, 2013) reported that cibarial pumping frequency is affected mainly by food viscosity rather than food sweetness.

We have cited Miyazaki et al. (J. Comp. Neurol, 2010), which shows the taste peg mechanosensory axons and the taste peg and taste bristle chemosensory axons project to distinct regions of the SEZ. Please see page 11.

Comment #3

A reference is needed for "NOMPC is reportedly expressed in labellar mechanosensory neurons" and for "the reverse may also be true, as is the case in the chordotonal neurons". The authors suggest that NOMPC might function with NAN, while *nompC* mutant flies show normal discrimination of agar softness. How can this be explained?

We added references to the sentence beginning, "NompC's expression in the labellar taste bristle MNs..." (Walker et al., Science 2000; Liang et al., Curr Biol 2013). We also added references to the sentence including "the reverse may also be true, as is the case in the chordotonal neurons" (Lehnert et al., Neuron 2013; Zhang et al., PNAS 2013). These are on page 12.

NOMPC has been implicated in *Drosophila* mechanosensation, but its exact role is somewhat controversial. Yan et al. and Liang et al. suggest NOMPC is the pore-forming ion channel subunit in the multidendritic neurons of the larval body wall and the direct mechanosensor in campaniform sensilla, respectively (Yan Z et al., Nature 2013; Liang X et al Curr Biol. 2013). Lehnert et al., on the other hand, argues Nan and lav are the direct mechanosensor and NOMPC is a mechanical signal modifier in adult chordotonal organs (Lehnert et al., Neuron 2013).

Labellar mechanosensory neurons express both Nan and NOMPC. It is possible Nan is the primary mechanosensor and NOMPC is a modifier or *vice versa*. Nevertheless, loss of *nan* usually causes a more severe defect than loss of *Nompc* mutations with regard to mechanosensation. Our current behavioral assay is not as sensitive as electrophysiology or calcium imaging, so we were unable to exclude the possibility that NOMPC is involved in food hardness detection.

Page 8, line 8,
PIEZO should be "Piezo".

We have corrected this.

Comment #4

[Redacted]

Table S1.

More explanation is needed. The numbers shown here must be no. of GFP-positive mechanosensory cells and what is the total number of mechanosensory cells in the labellum and in the taste peg? It should be important to know what percentage of cells is GFP-positive.

As requested, we have added the number of mechanosensory neurons and added details to the legend of Supplementary Table 1.

There are 31 mechanosensory neurons associated with chemosensory bristles and 2 mechanosensory neurons in lateral mechanosensilla per half of the labellum (Stocker et al., Cell Tissue Res 1994). Falk et al. observed that each taste peg houses one chemosensory and one mechanosensory neuron (J Morphol, 1976). The number of taste pegs per half of the labellum, however, varies from 24–42 depending on the reports (Shanbhag et al., Cell Tissue Res 2001, Falk et al., J Morphol, 1976). In our own observations, *Drosophila* labella average 30 taste pegs each hemisphere. This suggests the number of mechanosensory neurons per half of the labellum is also 30. Together, this would put the total number of mechanosensory neurons per half of labellum at 63.

Reviewer #2:

The revised manuscript has improved but failed to address one critical concern. The experiments suggesting that the mechanosensory neurons are GABAergic and inhibit sugar neurons are not well-executed and none of the suggested controls were done.

1) There is no evidence that any mechanosensory neuron in *Drosophila* is GABAergic. This would need to be demonstrated by immunostaining. (GAD-Gal4 may also be acceptable)

We used the vesicular GABA transporter reporter (*Vgat-GAL4*) to drive the membrane-tethered *UAS-mCD8::GFP* reporter and found GFP expression in labellar mechanosensory neurons. This is now described in the main text on page 10 and presented in the new Supplementary Fig. 4.

“We were able to confirm the labellar MNs are GABAergic by visualizing their expression of a *Vgat-GAL4*-driven *UAS-mCD8::GFP* reporter (Supplementary Fig. 4e). The GFP-labeled neurons showed typical MN morphology (Supplementary Fig. 4f).”

2) There is a lot of evidence that GABA feeds back onto sugar cells to inhibit their activity through GABA interneurons. It seems plausible that a similar mechanism operates here. As the R41E11 line is expressed in many neurons in the SEZ, it is possible that GABA RNAi is impacting interneurons rather than mechanosensory neurons.

We have now compared the levels of *Gad1* and *Vgat* between control and

GABAergic component-depleted labella using qPCR. Using *R41E11-GAL4* to drive each RNAi transgene, reduces the labellar levels of *Gad1* and *Vgat* but does not affect the expression of *nan*. These data are now included in the new Supplementary Figure 4.

“We quantified the knockdown efficiency and specificity of the *Vgat* and *Gad1* RNAi lines using qPCR (Supplementary Fig. 4b-d).

If GABAergic interneurons are involved in hardness-dependent food preference, they likely receive cholinergic innervation from the mechanosensory neurons and affect the sweet GRNs. Knock-down of cholinergic components (*Cha*, *VChaT*) in the mechanosensory neurons, however, did not affect hardness-dependent food preference. This suggests the mechanosensory neurons directly interact with the sweet GRNs.

My strong recommendation would be either

- A) that the authors show that the mechanosensory neurons are GABAergic and enzymes are knocked down in mechanosensory neurons
- B) or that they eliminate Fig 5A and B until they can demonstrate the existence of GABA mechanosensory neurons.

REVIEWERS' COMMENTS:

Reviewer #1 (Remarks to the Author):

The authors faithfully responded to all my comments and modified their manuscript accordingly.

Reviewer #2 (Remarks to the Author):

The modified manuscript has addressed my previous concerns and is appropriate for publication.

Point-by-point response

Both reviewers are satisfied with our revised manuscript.
Please see below.

Reviewer #1 (Remarks to the Author):

The authors faithfully responded to all my comments and modified their manuscript accordingly.

Reviewer #2 (Remarks to the Author):

The modified manuscript has addressed my previous concerns and is appropriate for publication.